# Recombinant zoster vaccine is associated with a reduced risk of dementia

Emily Rayens [1] ✉, Lina S. Sy [1], Lei Qian [1], Bradley K. Ackerson [1],
Julia Tubert[1], Yi Luo [1], Punam P. Modha[1], Raul O. Calderon [1],
Elizabeth Chmielewski-Yee [2], Driss Oraichi[2], Huifeng Yun[2], Carol Koro[3,4] &
Hung Fu Tseng [1,5]

Vaccines to prevent herpes zoster have been associated with reduced dementia risk. We conducted a retrospective matched cohort study of Kaiser Permanente Southern California members aged ≥65 years who received two doses of RZV 4 weeks–6 months apart between 01 April 2018 and 31 December 2020, with no dementia diagnoses or dementia medications prior to or within 6 months of their second RZV dose. Cox regression with inverse probability of treatment weighting was used to estimate adjusted hazard ratios (aHRs). The study included 65,800 RZV-vaccinated individuals and 263,200 unvaccinated matches. Vaccination with two doses of RZV was associated with a 51% lower risk of dementia (aHR: 0.49 [95% confidence interval (CI): 0.46–0.51]); aHRs were comparable across age, and racial and ethnic groups, but the risk reduction was stronger in females compared to males. In an evaluation of potential healthy vaccinee bias, the aHR of dementia for RZV compared to Tdap was 0.73 (95% CI: 0.67–0.79). Vaccination with two doses of RZV is associated with a statistically significant reduction in the risk of dementia in adults aged ≥65 years. After accounting for healthy vaccinee bias, RZV vaccination remains associated with a statistically significant lower risk of dementia.

Dementia is a major public health issue worldwide and one of the major causes of disability and dependency among older people[1]. Dementia can be caused by various conditions that gradually destroy nerve cells and damage the brain, leading to cognitive impairment. Alzheimer's disease is the most common form of dementia, present in an estimated 60–80% of dementia cases[2]. The number of people living with dementia worldwide has been increasing due to population growth and aging. In 2019, approximately 57.4 million people were living with dementia globally, and it has been estimated that this number will increase to approximately 152.8 million by 2050[3]. In 2024, in the United States (US), approximately 6.9 million adults aged 65 years and older were living with dementia[2].

A potential role of varicella-zoster virus (VZV) in the development of dementia has been considered[4–9]. Most people acquire VZV in childhood; the primary infection causes varicella (chickenpox), but the virus remains latent in the peripheral nervous system throughout life, and, upon reactivation, causes herpes zoster (HZ; commonly known as shingles), which manifests as a painful vesicular rash[10,11]. Several studies have shown an association between HZ and increased risk of dementia[4,12–16]. However, the exact mechanism by which viral reactivation leads to development of dementia has not been clearly demonstrated. Potential mechanisms may include neurodegeneration in the brain through neuroinflammation and/or vasculopathy, or via reactivation, possibly subclinical, of other viruses in the brain, such as herpes simplex virus-1[17–20].

[1]Department of Research & Evaluation, Kaiser Permanente Southern California, Pasadena, CA, USA. [2]GSK, Rockville, MD, USA. [3]GSK, Collegeville, PA, USA. [4]Department of Pharmaceutical Health Services Research, School of Pharmacy, University of Maryland, Baltimore, MD, USA. [5]Department of Health Systems Science, Kaiser Permanente Bernard J. Tyson School of Medicine, Pasadena, CA, USA. ✉e-mail: Emily.X.Rayens@kp.org

Recent studies with a live-attenuated HZ vaccine (zoster vaccine live [ZVL; *Zostavax*, Merck]) provided real-world evidence that immunization to prevent HZ may offer a protective effect against dementia[21–24]. However, these studies mostly compared the occurrence of dementia among HZ-vaccinated versus unvaccinated individuals, a design which may not account for a "healthy vaccinee" bias[25]. However, a recently published natural experiment in Wales using the SAIL Databank[26] compared adults who were immediately born before and after the eligibility cut-off date to receive ZVL, ensuring the greatest possible natural balance of potential confounding variables[27]. This study demonstrated that receiving ZVL led to a 20% relative reduction in probability of a new dementia diagnosis over a follow-up period of 7 years compared to those who did not receive ZVL[27].

To date, most studies on HZ vaccination and dementia have primarily examined ZVL, which was approved by the US Food and Drug Administration (FDA) in 2006[28] and discontinued in the US in 2020[29]. In 2017, the recombinant zoster vaccine (RZV; *Shingrix*, GSK) received FDA approval and was preferentially recommended by the Advisory Committee on Immunization Practices (ACIP) for all adults ≥50 years of age, even those who had previously received ZVL, as RZV was demonstrated to have greater efficacy and durability[30–33]. Given the relatively recent approval and implementation of RZV vaccination, research on the relationship between RZV vaccination and dementia risk is limited.

A recent natural experiment of dementia risk in the US compared adults who received RZV between 2017 and 2020 to those who received ZVL between 2014 and 2017, using data from the TriNetX US Collaborative Network[34]. In this study, RZV was associated with a 17% increase in time free from dementia diagnosis, but the study lacked validation of diagnoses and had limited information on socioeconomic factors[34]. Another claims-based study using data from the Optum Labs Data Warehouse examined RZV and dementia using a retrospective cohort design and found that patients who received either two doses or one dose of RZV had a statistically significant reduced risk of dementia (adjusted hazard ratio [aHR] 0.68 and 0.89, respectively) as compared to unvaccinated individuals[35].

In our current study conducted at Kaiser Permanente Southern California (KPSC), we evaluated the potential association between immunization with two doses of RZV and reduction in risk of dementia in adults aged 65 years and older in the US, while addressing several limitations of previous studies. An overview of the study objectives and design can be found in Fig. 1a–c.

## Results

### Identification of dementia and mild cognitive impairment outcomes

To assess the quality of dementia diagnosis codes, 100 patient records with ≥1 coded dementia diagnosis (primary definition) were randomly selected for chart review of the outcome. Of the 100 cases reviewed, 94 were confirmed by chart review to have clinical documentation to support the coded diagnosis (positive predictive value [PPV]: 94.0% [95% confidence interval (CI): 87.4–97.8%]; Supplementary Table 1). Subsets of the 100 records randomly selected with the primary dementia definition also met alternative dementia definitions and were used to calculate the associated PPV. The PPV of the alternative dementia definitions were 95.9% (95% CI: 88.6–99.2%) for ≥2 dementia diagnoses, 97.0% (95% CI: 84.2–99.9%) for ≥1 dementia diagnosis with dementia medication, and 96.1% (95% CI: 88.9–99.2%) for either ≥2 dementia diagnoses or ≥1 dementia diagnosis with dementia medication (Supplementary Table 1). Of the 100 randomly selected records with coded mild cognitive impairment (MCI) diagnosis that were chart reviewed, 98 were confirmed (PPV: 98.0% [95% CI: 93.0–99.8%]).

### Dementia in RZV-vaccinated versus unvaccinated individuals

There were 65,800 two-dose RZV-vaccinated, and 263,200 matched unvaccinated individuals included in the analysis examining the association between two doses of RZV and risk of dementia (Fig. 1d and Table 1). Compared to unvaccinated individuals, a higher proportion of RZV-vaccinated individuals had a recent history of HZ (≤2 years), receipt of influenza vaccines and other vaccines in the year prior to the index date (the date of receipt of the second dose of RZV; matched unvaccinated individuals were assigned the same index date), preventive care and outpatient or virtual visits in the year prior to the index date, and longer continuous KPSC membership prior to the index date (≥10 years) (absolute standard difference [ASD] >0.1 before weighting; Supplementary Table 2). Vaccinated individuals had a higher frequency of dyslipidemia diagnosis, were less likely to be smokers, had lower body mass index (BMI), and had higher neighborhood-level income and education as compared to unvaccinated individuals (ASD > 0.1 before weighting). The mean follow-up time (starting 6 months after the index date) was 3.40 (standard deviation [SD]: 0.99) years for RZV-vaccinated individuals and 1.81 (SD: 1.27) years for unvaccinated individuals (Supplementary Table 2). The mean follow-up time in unvaccinated individuals was shorter, as individuals were censored if they received RZV. After weighting, all baseline characteristics were well-balanced between vaccinated and unvaccinated cohorts (Supplementary Fig. 1). After weighting, the mean age was 73.42 (SD: 6.19) years and 73.33 (SD: 6.69) years in the RZV-vaccinated and unvaccinated populations, respectively. The majority of RZV-vaccinated and unvaccinated individuals were female (57.5% vs 57.7%, respectively) and non-Hispanic White (61.1% vs 61.2%, respectively; Table 1).

Among RZV-vaccinated individuals, there were 2401 diagnosed dementia cases during follow-up with an incidence rate (IR) of 10.74 (95% CI: 10.32–11.18) per 1000 person-years (PY) compared to 10,983 diagnosed dementia cases with an IR of 23.04 (95% CI: 22.61–23.48) per 1000 PY in the unvaccinated cohort (Fig. 2, Supplementary Table 3). The cumulative incidence of dementia was significantly higher in the unvaccinated cohort (10.64%) compared to the RZV-vaccinated cohort (5.67%; Fig. 3). The overall aHR for dementia comparing RZV-vaccinated to unvaccinated individuals was 0.49 (95% CI: 0.46–0.51). The aHRs for dementia were similar across age groups, racial and ethnic groups, history of MCI diagnosis, history of ZVL vaccination, and diagnosed dementia types (Alzheimer's disease and vascular dementia). In females, aHR was lower than that observed in males (0.45 [95% CI: 0.42–0.48] vs 0.55 [95% CI: 0.51–0.59]; Fig. 2 and Supplementary Table 3).

To evaluate incidence of dementia in the context of diagnosed HZ, RZV-vaccinated and unvaccinated individuals were stratified in an unadjusted post-hoc analysis by HZ status. In individuals without any diagnosed HZ, the IR per 1000 PY of dementia was 10.06 (95% CI: 9.62–10.53) and 22.29 (95% CI: 21.83–22.75) in RZV-vaccinated and unvaccinated adults, respectively (Supplementary Table 4). In individuals with diagnosed HZ, whether prior to or during follow-up, the IR of dementia per 1000 PY was 14.49 (95% CI: 13.27–15.82) and 27.87 (95% CI: 26.61–29.20) in RZV-vaccinated and unvaccinated adults, respectively (Supplementary Table 4).

### Dementia in RZV-vaccinated versus Tdap-vaccinated individuals

To evaluate potential healthy vaccinee bias, we compared dementia incidence in RZV-vaccinated individuals versus a separate comparator group who received tetanus, diphtheria, and acellular pertussis (Tdap) vaccine but not RZV. Tdap-vaccinated individuals were matched 1:1 to RZV-vaccinated individuals on age and index (vaccination) date in the same year or plus or minus 1 year if no matches were available in the same year. Individuals in the RZV-vaccinated cohort were permitted to also receive Tdap before or after the index date. This analysis included 65,800 two-dose RZV-vaccinated individuals (same cohort as the primary analysis) and 65,800 Tdap-vaccinated individuals (Supplementary Table 5). Compared to Tdap-vaccinated individuals, a higher proportion of RZV-vaccinated individuals were non-Hispanic White,

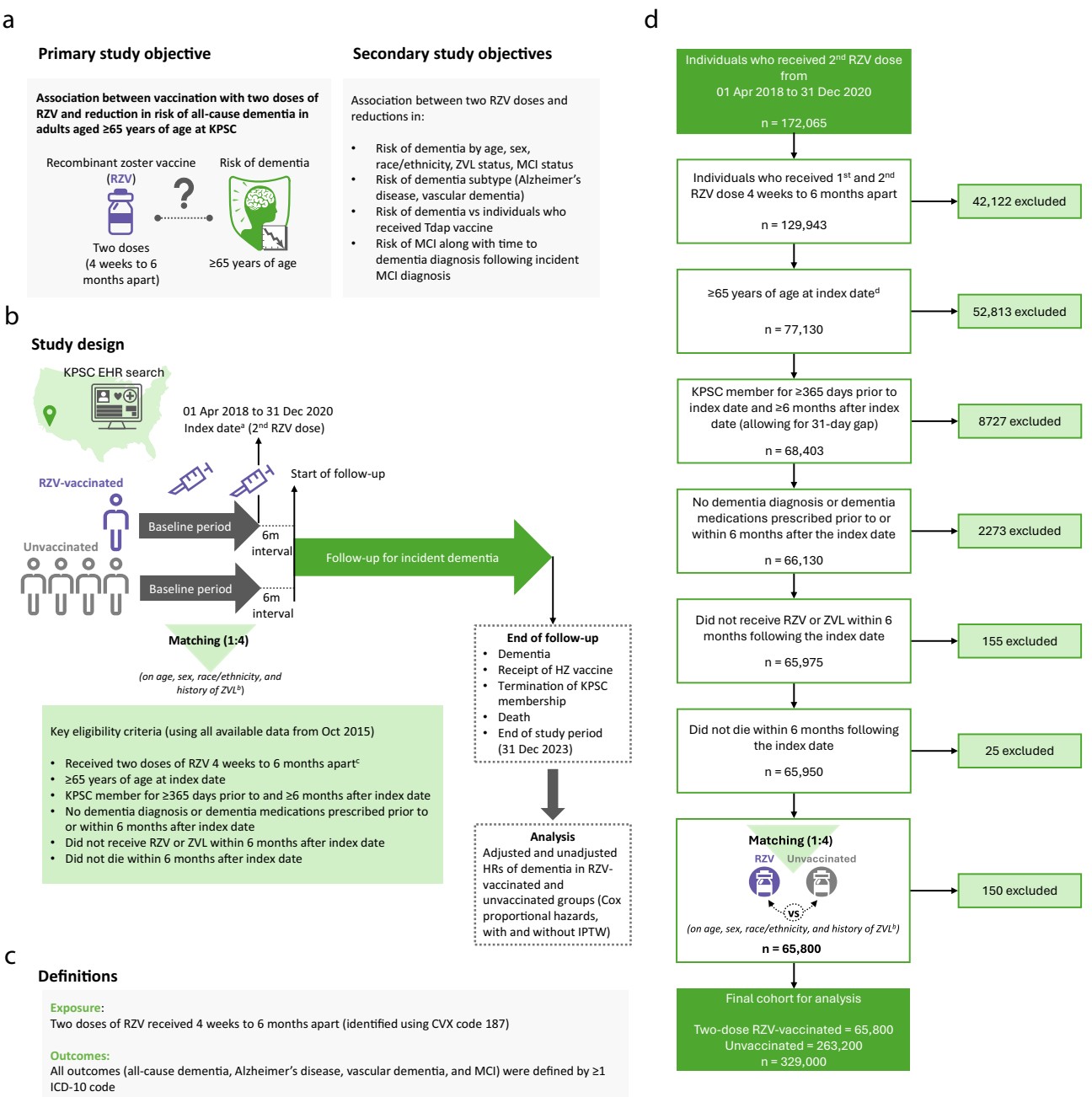

**Fig. 1 | Study objectives, study design and flow chart.** [a]Unvaccinated individuals were assigned the same index date as their vaccinated matches. [b]During the matching process, the following inclusion criteria were applied to unvaccinated comparators based on their assigned index date: no receipt of RZV prior to or within 6 months following the index date, no receipt of ZVL within 6 months following the index date, ≥65 years of age at index date, KPSC member for ≥365 days prior to index date and ≥6 months after index date (allowing for 31-day gap), no dementia diagnosis or dementia medications prescribed prior to or within 6 months of index date, and no death within 6 months following the index date.

[c]Only for the RZV-vaccinated group. [d]Index date for two-dose RZV-vaccinated recipients was the date of their second RZV dose. For unvaccinated comparators, index date was assigned based on the second RZV dose date of their matched counterpart. EHR electronic health record, HR hazard ratio, HZ herpes zoster, ICD-10 International Classification of Diseases, Tenth Revision, IPTW inverse probability of treatment weighting, KPSC Kaiser Permanente Southern California, m month, MCI mild cognitive impairment, n number, RZV recombinant zoster vaccine, Tdap tetanus, diphtheria and acellular pertussis vaccine, ZVL zoster vaccine live.

had a history of ZVL vaccination, recent history of HZ (≤2 years), receipt of influenza vaccine and other vaccines in the year prior to the index date, preventive care in the year prior to the index date, more outpatient and virtual visits, and longer continuous KPSC membership (≥10 years; ASD > 0.1 before weighting; Supplementary Table 5). Similar to the main analysis, RZV-vaccinated individuals had a higher frequency of dyslipidemia, were less likely to be smokers, had lower BMI, and had higher neighborhood-level income and education

compared to Tdap-vaccinated individuals (ASD > 0.1 before weighting; Supplementary Table 5). A higher proportion of Tdap-vaccinated individuals had diabetes, metabolic syndrome, a higher frailty index score, Medicaid coverage, more Emergency Department (ED) visits and hospitalizations, and earlier index dates (ASD > 0.1 before weighting; Supplementary Table 5). All covariates were well-balanced after weighting (Supplementary Fig. 2). The mean follow-up time was 3.40 years (SD: 0.99) for RZV-vaccinated individuals and 2.38 years

**Table 1 | Baseline characteristics of the two-dose RZV-vaccinated and unvaccinated cohorts after weighting**

| | Vaccinated n = 65,800 | Unvaccinated n = 263,200 | ASD |
|---|---|---|---|
| Age at index date, years | | | 0.014 |
| Mean (SD) | 73.42 (6.19) | 73.33 (6.69) | |
| Median | 72 | 72 | |
| Q1, Q3 | 69, 77 | 68, 77 | |
| Minimum, Maximum | 65, 101 | 65, 109 | |
| Age at index date, years, n (%) | | | 0.031 |
| 65–69 | 20,515 (31.2) | 85,322 (32.4) | |
| 70–74 | 20,796 (31.6) | 82,857 (31.5) | |
| 75–79 | 13,349 (20.3) | 52,606 (20.0) | |
| ≥80 | 11,140 (16.9) | 42,415 (16.1) | |
| Sex, n (%) | | | 0.004 |
| Female | 37,818 (57.5) | 151,775 (57.7) | |
| Male | 27,982 (42.5) | 111,425 (42.3) | |
| Race/Ethnicity, n (%) | | | 0.011 |
| Non-Hispanic White | 40,172 (61.1) | 161,188 (61.2) | |
| Non-Hispanic Black | 3052 (4.6) | 11,995 (4.6) | |
| Hispanic | 10,331 (15.7) | 41,506 (15.8) | |
| Non-Hispanic Asian | 10,724 (16.3) | 42,137 (16.0) | |
| Other/Multiple/Unknown | 1521 (2.3) | 6374 (2.4) | |
| History of ZVL vaccination[a], n (%) | | | 0.044 |
| No | 20,751 (31.5) | 88,250 (33.5) | |
| Yes, ≤5 years | 6093 (9.3) | 22,933 (8.7) | |
| Yes, >5 years | 38,956 (59.2) | 152,017 (57.8) | |
| MCI status[b], n (%) | | | 0.001 |
| No | 64,576 (98.1) | 258,352 (98.2) | |
| Yes | 1224 (1.9) | 4848 (1.8) | |
| History of HZ[a], n (%) | | | 0.007 |
| No | 57,260 (87.0) | 229,508 (87.2) | |
| Yes, ≤2 years | 2083 (3.2) | 8052 (3.1) | |
| Yes, >2 years | 6456 (9.8) | 25,641 (9.7) | |
| Length of continuous membership[a], years, n (%) | | | 0.011 |
| 1– <5 | 9490 (14.4) | 38,476 (14.6) | |
| 5– <10 | 11,466 (17.4) | 46,678 (17.7) | |
| ≥10 | 44,844 (68.2) | 178,046 (67.6) | |
| Other vaccines[c], n (%) | | | |
| Influenza vaccine | 55,343 (84.1) | 215,884 (82.0) | 0.056 |
| COVID-19 vaccine | 2 (0.0) | 18 (0.0) | 0.006 |
| Other[d] | 16,792 (25.5) | 63,237 (24.0) | 0.035 |
| Number of outpatient and virtual visits[c], n (%) | | | 0.059 |
| 0–4 | 12,001 (18.2) | 54,101 (20.6) | |
| 5–10 | 21,178 (32.2) | 83,255 (31.6) | |
| ≥11 | 32,621 (49.6) | 125,843 (47.8) | |
| Number of emergency department visits[c], n (%) | | | 0.014 |
| 0 | 52,364 (79.6) | 210,861 (80.1) | |
| 1 | 9093 (13.8) | 35,590 (13.5) | |
| ≥2 | 4343 (6.6) | 16,749 (6.4) | |
| Number of hospitalizations[c], n (%) | | | 0.005 |
| 0 | 61,097 (92.9) | 244,701 (93.0) | |
| 1 | 3525 (5.4) | 13,875 (5.3) | |
| ≥2 | 1178 (1.8) | 4624 (1.8) | |

**Table 1 (continued) | Baseline characteristics of the two-dose RZV-vaccinated and unvaccinated cohorts after weighting**

| | Vaccinated n = 65,800 | Unvaccinated n = 263,200 | ASD |
|---|---|---|---|
| Preventive care[c], n (%) | 37,467 (56.9) | 145,919 (55.4) | 0.030 |
| Comorbidities[c], n(%) | | | |
| Kidney disease | 11,801 (17.9) | 45,606 (17.3) | 0.016 |
| Heart disease | 5933 (9.0) | 22,878 (8.7) | 0.011 |
| Lung disease | 11,910 (18.1) | 46,080 (17.5) | 0.016 |
| Liver disease | 2911 (4.4) | 11,210 (4.3) | 0.008 |
| Diabetes | 18,136 (27.6) | 70,108 (26.6) | 0.021 |
| Brain tumor | 49 (0.1) | 232 (0.1) | 0.005 |
| Traumatic brain injury | 289 (0.4) | 1079 (0.4) | 0.005 |
| Hearing loss | 4653 (7.1) | 17,946 (6.8) | 0.010 |
| Parkinson's disease | 524 (0.8) | 2073 (0.8) | 0.001 |
| Huntington's disease | 3 (0.0) | 13 (0.0) | 0.000 |
| Multiple sclerosis | 125 (0.2) | 496 (0.2) | 0.000 |
| Anxiety | 7237 (11.0) | 28,188 (10.7) | 0.009 |
| Depression | 9061 (13.8) | 35,379 (13.4) | 0.010 |
| Sleep disorders | 9119 (13.9) | 35,188 (13.4) | 0.014 |
| Metabolic syndrome | 6529 (9.9) | 25,164 (9.6) | 0.012 |
| Hypertension | 38,803 (59.0) | 150,217 (57.1) | 0.039 |
| Alcohol abuse | 992 (1.5) | 3990 (1.5) | 0.001 |
| Dyslipidemia | 45,405 (69.0) | 175,752 (66.8) | 0.048 |
| Viral infections | 2599 (3.9) | 10,310 (3.9) | 0.002 |
| Immunocompromised at index date[c], n (%) | 3234 (4.9) | 12,724 (4.8) | 0.004 |
| Charlson comorbidity score[c,f] | | | 0.017 |
| Mean (SD) | 1.80 (2.04) | 1.76 (2.08) | |
| Median | 1 | 1 | |
| Q1, Q3 | 0, 3 | 0, 3 | |
| Minimum, maximum | 0, 17 | 0, 17 | |
| Charlson comorbidity score[c,f], n (%) | | | 0.043 |
| 0 | 22,480 (34.2) | 95,296 (36.2) | |
| 1 | 14,395 (21.9) | 56,214 (21.4) | |
| ≥2 | 28,925 (44.0) | 111,690 (42.4) | |
| Frailty index[c,g] | | | 0.009 |
| Mean (SD) | 0.14 (0.04) | 0.14 (0.04) | |
| Median | 0.13 | 0.13 | |
| Q1, Q3 | 0.11, 0.16 | 0.11, 0.15 | |
| Minimum, maximum | 0.04, 0.37 | 0.05, 0.40 | |
| Frailty index[c,g], n (%) | | | 0.042 |
| Q1 | 15,273 (23.2) | 65,742 (25.0) | |
| Q2 | 16,718 (25.4) | 65,816 (25.0) | |
| Q3 | 16,827 (25.6) | 65,812 (25.0) | |
| Q4, most frail | 16,982 (25.8) | 65,830 (25.0) | |
| Smoking[c], n(%) | | | 0.077 |
| No | 44,869 (68.2) | 175,742 (66.8) | |
| Yes | 16,940 (25.7) | 66,288 (25.2) | |
| Unknown | 3991 (6.1) | 21,169 (8.0) | |
| Body mass index[h], n (%) | | | 0.081 |
| <18.5 | 934 (1.4) | 3819 (1.5) | |
| 18.5– <25 | 18,993 (28.9) | 74,294 (28.2) | |
| 25– <30 | 22,905 (34.8) | 89,528 (34.0) | |
| ≥30 | 18,832 (28.6) | 73,446 (27.9) | |

**Table 1 (continued) | Baseline characteristics of the two-dose RZV-vaccinated and unvaccinated cohorts after weighting**

| | Vaccinated<br>n = 65,800 | Unvaccinated<br>n = 263,200 | ASD |
|---|---|---|---|
| Unknown | 4136 (6.3) | 22,113 (8.4) | |
| Neighborhood median household income, n (%) | | | 0.004 |
| <$40,000 | 2825 (4.3) | 11,384 (4.3) | |
| $40,000–$59,999 | 10,700 (16.3) | 43,016 (16.3) | |
| $60,000–$79,999 | 14,508 (22.0) | 58,059 (22.1) | |
| ≥$80,000 | 37,610 (57.2) | 150,081 (57.0) | |
| Unknown | 157 (0.2) | 661 (0.3) | |
| Neighborhood-level education[i], n (%) | | | 0.007 |
| ≤High school | 12,522 (19.0) | 50,807 (19.3) | |
| >High school | 53,123 (80.7) | 211,742 (80.4) | |
| Unknown | 155 (0.2) | 651 (0.2) | |
| Medicaid, n (%) | 2487 (3.8) | 9637 (3.7) | 0.006 |
| Year of index date, n (%) | | | 0.029 |
| 2018 | 8566 (13.0) | 32,688 (12.4) | |
| 2019 | 35,392 (53.8) | 139,812 (53.1) | |
| 2020 | 21,842 (33.2) | 90,700 (34.5) | |

*AIDS* acquired immunodeficiency syndrome, *ASD* absolute standardized difference, *HIV* human immunodeficiency virus, *HZ* herpes zoster, *MCI* mild cognitive impairment, *n* number, *Q* quartile, *RZV* recombinant zoster vaccine, *SD* standard deviation, *Td* tetanus and diphtheria vaccine, *Tdap* tetanus, diphtheria, and acellular pertussis vaccine, *ZVL* zoster vaccine live.
[a]Defined based on all available medical records prior to the index date.
[b]Defined based on all available medical records prior to index date to 6 months after the index date.
[c]Defined in the year prior to the index date.
[d]Among subjects who received other vaccines in the year prior to the index date: pneumococcal (60.2%), Tdap/Td (31.8%), hepatitis A or B (10.1%), and other vaccine (13.0%).
[e]Immunocompromised defined as HIV/AIDS, hematopoietic stem cell/solid organ transplant, leukemia/lymphoma, congenital and other immunodeficiencies, or asplenia/hyposplenia at any time prior to the index date, or immunosuppressive medication at index date.
[f]Possible range: 0–29[46].
[g]Possible range: 0–1[47].
[h]Defined as most recent in the year prior to the index date.
[i]Defined as <50% or ≥50% of the neighborhood that attained more than high school education.

(SD: 1.27) for individuals in the Tdap comparator group. Most RZV-vaccinated individuals (98.6%) had also received Tdap by the end of follow-up. There were 2118 Tdap-vaccinated individuals (3.2%) who contributed person-time to both cohorts as they were vaccinated with RZV during follow-up (Supplementary Table 5). After weighting, the populations were similar. The mean ages were 73.34 (SD: 6.32) years and 73.23 (SD: 6.29) years in the RZV-vaccinated and Tdap-vaccinated groups, respectively. In the RZV-vaccinated group, 55.5% were female and 53.7% were non-Hispanic White, compared to 55.7% female and 53.4% non-Hispanic White in the Tdap group (Supplementary Table 6).

Among RZV-vaccinated individuals, there were 2401 dementia cases during follow-up with an IR of 10.74 (95% CI: 10.32–11.18) per 1000 PY compared to 2835 dementia cases with an IR of 18.09 (95% CI: 17.44–18.77) per 1000 PY in the Tdap-vaccinated cohort (Fig. 4 and Supplementary Table 7). The cumulative incidence of dementia was statistically significantly higher in the Tdap-vaccinated cohort (9.90%) compared to the RZV-vaccinated cohort (5.67%; Fig. 5). The aHR of dementia in RZV-vaccinated individuals compared to Tdap-vaccinated individuals was 0.73 (95% CI: 0.67–0.79; Fig. 4 and Supplementary Table 7).

## Negative control outcomes in RZV-vaccinated versus unvaccinated individuals

To evaluate potential residual confounding, we assessed the relationship between RZV and a composite negative control outcome (NCO) in the RZV-vaccinated and unvaccinated cohorts in a post-hoc analysis (Supplementary Table 8 and Fig. 6). Overall, there were 2171 NCO cases reported in the RZV-vaccinated cohort (IR: 9.75 [95% CI: 9.34–10.16] per 1000 PY) and 4877 in the unvaccinated cohort (IR: 10.06 [95% CI: 9.78–10.35] per 1000 PY). The overall aHR of two doses of RZV and the composite NCO was 0.94 (95% CI: 0.89–1.00; Fig. 6 and Supplementary Table 8).

## E-value

The E-value for the primary analysis result was 3.50 for the point estimate and 3.33 for the upper bound of the CI.

## MCI in RZV-vaccinated versus unvaccinated individuals

The analysis to evaluate the association between RZV and risk of MCI included 64,744 two-dose RZV-vaccinated and 258,199 unvaccinated individuals with no history of MCI prior to follow-up (Supplementary Table 9). The cohort used for analysis of MCI outcomes was a subset of that used in analysis of the primary dementia outcome, differing only on exclusion of individuals with MCI diagnosis prior to follow-up. As such, the baseline characteristics of the RZV-vaccinated and unvaccinated populations are similar to those reported above (Table 1 and Supplementary Tables 9–10). For the MCI analysis, the mean follow-up time was 3.41 (SD: 0.98) years for RZV-vaccinated individuals and 1.85 (SD: 1.28) years for unvaccinated individuals (Supplementary Table 9).

There were 1557 diagnosed MCI cases among RZV-vaccinated individuals (IR: 7.05 [95% CI: 6.71–7.41] per 1000 PY) compared to 3564 diagnosed MCI cases among unvaccinated individuals (IR: 7.47 [95% CI: 7.23–7.72] per 1000 PY; Fig. 7, Supplementary Table 11). The cumulative incidence of MCI was significantly higher in the unvaccinated cohort (Supplementary Fig. 3). Due to violation of the proportional hazards assumption, the analysis was stratified into <3.5 years of follow-up and ≥3.5 years of follow-up. The aHR of MCI in RZV-vaccinated individuals compared to unvaccinated individuals with follow-up of <3.5 years was 0.84 (95% CI: 0.78–0.90; Fig. 7 and Supplementary Table 11). The number of MCI cases after ≥3.5 years of follow-up was small.

There were 393 RZV-vaccinated and 909 unvaccinated individuals who were diagnosed with incident MCI and subsequently dementia during the follow-up period. The median time (interquartile range [IQR]) between MCI and dementia diagnosis was 261 (IQR: 98, 500) days and 193 (IQR: 37, 396) days in the RZV-vaccinated and unvaccinated cohorts, respectively (Supplementary Table 12).

## Sensitivity analyses

In consideration of the COVID-19 pandemic and its effect on healthcare seeking behaviors, we conducted a sensitivity analysis excluding RZV-vaccinated individuals and their unvaccinated matches with an index date from 01 March 2020 through 31 December 2020. In this cohort, there were 50,542 RZV-vaccinated individuals and 202,168 unvaccinated individuals, with characteristics similar to the primary cohort (Table 1 and Supplementary Table 13). There were 2012 dementia cases (IR: 10.98 [95% CI: 10.51–11.47] per 1000 PY) in the RZV-vaccinated cohort and 9383 dementia cases (IR: 25.28 [95% CI: 24.77–25.80] per 1000 PY) in the unvaccinated cohort (Supplementary Table 14). The aHR of two doses of RZV and dementia in this alternative accrual period was 0.45 (95% CI: 0.43–0.48).

Using an alternative outcome definition for dementia (≥2 diagnoses or ≥1 diagnosis with medication; Supplementary Methods), there were 1757 dementia cases (IR: 7.84 [95% CI: 7.48–8.21] per 1000 PY) in the RZV-vaccinated cohort and 7,742 dementia cases (IR: 16.14 [95% CI: 15.79–16.51] per 1000 PY) in the unvaccinated cohort. The aHR of two doses of RZV and dementia using an alternative definition was 0.49 (95% CI: 0.46–0.52; Supplementary Table 14).

In a post-hoc sensitivity analysis in which individuals were not censored upon receipt of HZ vaccine during follow-up, there were

| | Incidence per 1000 person-years (95% CI) | | aHR (95% CI)[a] | |
|---|---|---|---|---|
| **Dementia** | **Vaccinated** | **Unvaccinated** | | |
| Overall | 10.74 (10.32–11.18) | 23.04 (22.61–23.48) | | 0.49 (0.46–0.51) |
| Dementia subtype | | | | |
| Alzheimer's disease | 3.17 (2.95–3.42) | 5.74 (5.53–5.95) | | 0.48 (0.44–0.53) |
| Vascular dementia | 1.04 (0.91–1.18) | 2.33 (2.20–2.47) | | 0.45 (0.39–0.53) |
| Age at index date, years | | | | |
| 65–69[b] | 2.59 (2.25–2.99) | 4.42 (4.10–4.76) | | 0.53 (0.44–0.64) |
| 70–79 | 8.85 (8.33–9.40) | 17.90 (17.38–18.43) | | 0.49 (0.45–0.53) |
| ≥80[c] | 34.33 (32.44–36.34) | 83.35 (81.21–85.56) | | 0.48 (0.45–0.52) |
| Sex | | | | |
| Female | 10.07 (9.54–10.62) | 23.82 (23.25–24.40) | | 0.45 (0.42–0.48) |
| Male | 11.68 (11.01–12.39) | 21.98 (21.33–22.64) | | 0.55 (0.51–0.59) |
| Race/Ethnicity, n (%) | | | | |
| Non-Hispanic White | 11.08 (10.54–11.64) | 25.10 (24.52–25.69) | | 0.46 (0.43–0.48) |
| Non-Hispanic Black[d] | 14.58 (12.38–17.16) | 28.96 (26.87–31.21) | | 0.49 (0.40–0.60) |
| Hispanic[e] | 10.53 (9.50–11.66) | 20.63 (19.66–21.65) | | 0.55 (0.48–0.63) |
| Non-Hispanic Asian[f] | 9.04 (8.11–10.07) | 17.98 (17.03–18.97) | | 0.51 (0.44–0.58) |
| ZVL status | | | | |
| No[g] | 9.94 (9.25–10.70) | 19.80 (19.17–20.45) | | 0.51 (0.47–0.56) |
| Yes | 11.12 (10.60–11.66) | 25.15 (24.57–25.73) | | 0.46 (0.43–0.49) |
| MCI status | | | | |
| No | 9.53 (9.14–9.95) | 20.16 (19.76–20.57) | | 0.48 (0.46–0.51) |
| Yes[h] | 99.00 (88.39–110.87) | 219.86 (208.31–232.06) | | 0.53 (0.47–0.61) |

0.0  0.2  0.4  0.6  0.8  1.0

**Fig. 2 | Risk of dementia among two-dose RZV-vaccinated versus unvaccinated individuals.** Incidence of dementia with 95% confidence intervals and adjusted hazard ratios with 95% confidence intervals calculated using a Cox regression model with IPTW are presented for two-dose RZV-vaccinated ($n = 65,800$) compared to unvaccinated ($n = 263,200$) individuals. [a]Adjusted for time-varying vaccination status (influenza, COVID-19, and other vaccines) besides IPTW. [b]Adjusted for age in addition to time-varying vaccination status and IPTW. [c]Adjusted for age and BMI in addition to time-varying vaccination status and IPTW. [d]More than 20% of covariates were imbalanced (ASD > 0.1) within the subgroup. Recalculated propensity scores within the subgroup for IPTW. [e]Adjusted for BMI, smoking, frailty index, dyslipidemia, number of outpatient and virtual visits, and Charlson comorbidity score in addition to time-varying vaccination status and IPTW. [f]Adjusted for BMI in addition to time-varying vaccination status and IPTW. [g]Adjusted for BMI, smoking, number of outpatient and virtual visits, and dyslipidemia in addition to time-varying vaccination status and IPTW. [h]Adjusted for age, BMI, smoking, number of outpatient and virtual visits, liver disease, and anxiety in addition to time-varying vaccination status and IPTW. aHR adjusted hazard ratio, ASD absolute standard difference, BMI body mass index, CI confidence interval, IPTW inverse probability of treatment weighting, MCI mild cognitive impairment, n number, RZV recombinant zoster vaccine, ZVL zoster vaccine live.

39,813 RZV-vaccinated individuals and 159,252 unvaccinated individuals (Supplementary Table 15), with characteristics similar to the primary cohort. There were 1540 dementia cases (IR: 10.95 [95% CI: 10.42–11.51] per 1000 PY) in the RZV-vaccinated cohort and 9834 dementia cases (IR: 20.02 [95% CI: 19.62–20.42] per 1000 PY) in the unvaccinated cohort (Supplementary Table 14). The aHR of two doses of RZV and dementia was 0.60 (95% CI: 0.56–0.63).

A final post-hoc sensitivity analysis was conducted in which the follow-up time began 1 year, instead of 6 months, after the index date. The number of dementia cases in the RZV-vaccinated cohort was 2186 (IR: 11.44 [95% CI: 10.97–11.93] per 1000 PY) compared to 7066 (IR: 19.27 [95% CI: 18.83–19.73] per 1000 PY) in the unvaccinated cohort and 2367 (IR: 18.93 [95% CI: 18.18–19.71] per 1000 PY) in the Tdap-vaccinated cohort. The aHR of two doses of RZV and dementia with follow-up starting 1 year after the index date was 0.60 (95% CI: 0.57–0.64) and 0.73 (95% CI: 0.67–0.79) compared to unvaccinated individuals and Tdap-vaccinated individuals, respectively (Supplementary Table 16). The cumulative incidence of dementia was significantly lower in the RZV-vaccinated group (5.36%) compared to the unvaccinated (9.09%; Supplementary Fig. 4) and Tdap-vaccinated (9.23%; Supplementary Fig. 5) groups.

## Discussion
In this study, two doses of RZV received 4 weeks to 6 months apart were associated with a statistically significant reduction in dementia risk in adults aged 65 years and older. Furthermore, the reduction in dementia risk was consistent across age groups, racial and ethnic groups, history of ZVL, history of MCI diagnosis, and dementia types, including Alzheimer's disease and vascular dementia, with the reduction in risk ranging from 45% to 55%.

In stratified analyses, we found that the reduction in dementia risk was greater in females compared to males (55% vs 45%, respectively). This finding was consistent with a recent natural experiment of dementia risk that compared adults who received RZV to those who received ZVL, which found that while the increased protective association with RZV was present in both males and females, the association was stronger in females[34]. In the natural experiment of ZVL immunization that occurred based on birth date cutoffs in eligibility, significant vaccine effects were only observed in females, beyond the overall 20% reduction in risk of dementia associated with ZVL vaccination[27].

Two doses of RZV were additionally associated with a reduction in risk of incident diagnosis of MCI. We further observed that RZV-

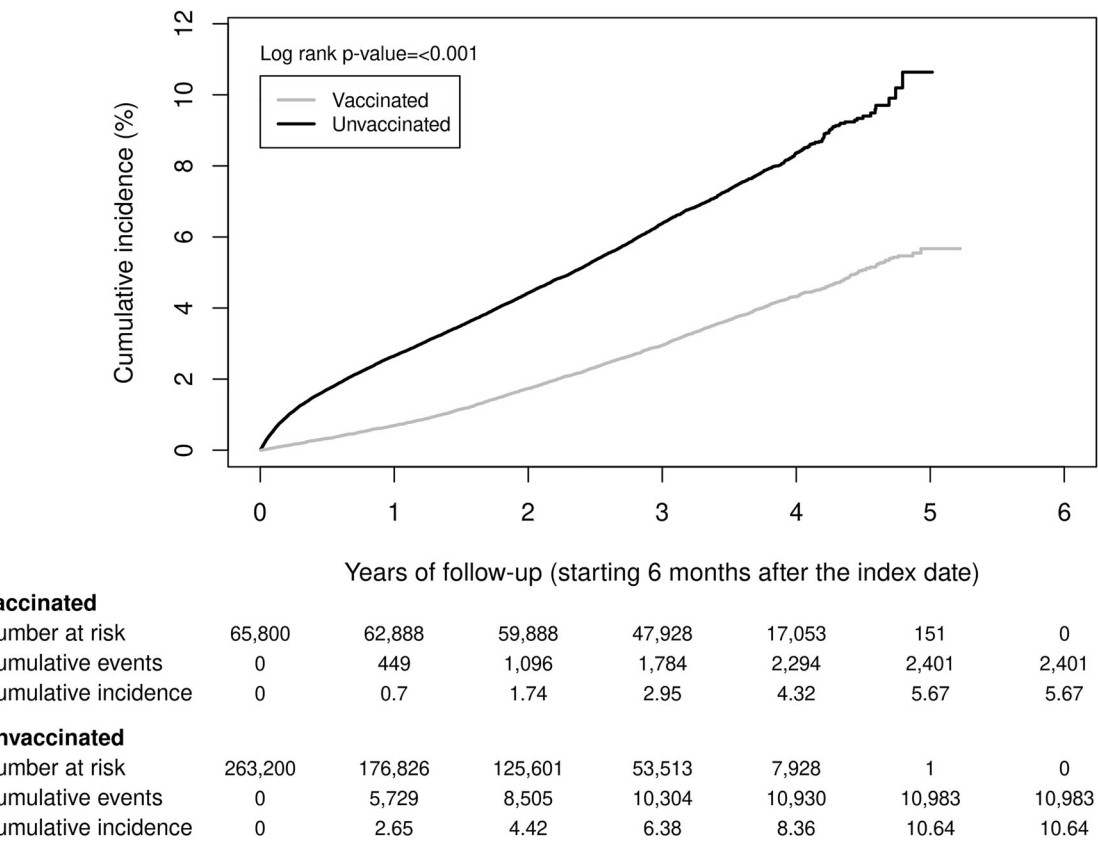

Fig. 3 | **Cumulative incidence of dementia among two-dose RZV-vaccinated versus unvaccinated individuals.** RZV recombinant zoster vaccine.

| Dementia | Incidence per 1000 person-years (95% CI) | | aHR (95% CI)[a] |
|---|---|---|---|
| | RZV-Vaccinated | Comparison Group | |
| RZV Comparison Group | | | |
| Unvaccinated | 10.74 (10.32–11.18) | 23.04 (22.61–23.48) | 0.49 (0.46–0.51) |
| Tdap | | 18.09 (17.44–18.77) | 0.73 (0.67–0.79) |

Fig. 4 | **Risk of dementia among two-dose RZV-vaccinated versus unvaccinated or Tdap-vaccinated individuals.** Incidence of dementia with 95% confidence intervals and adjusted hazard ratios with 95% confidence intervals calculated using a Cox regression model with IPTW are presented for two-dose RZV-vaccinated ($n = 65,800$) compared to Tdap-vaccinated ($n = 65,800$) and unvaccinated ($n = 263,200$) individuals. [a]Adjusted for time-varying vaccination status (influenza, COVID-19, and other vaccines) besides IPTW. aHR adjusted hazard ratio, CI confidence interval, IPTW inverse probability of treatment weighting, n number, RZV recombinant zoster vaccine, Tdap tetanus, diphtheria, and acellular pertussis vaccine.

vaccinated individuals had a greater length of time (median difference of 68 days) between diagnosis of MCI and diagnosis of dementia compared to unvaccinated individuals. It is of note that MCI was less diagnosed compared to dementia in this cohort, despite expectations that MCI would affect a larger portion of the adult population aged 65 years and older. However, it is estimated that up to 92% of MCI cases may go undiagnosed[36,37].

While the mechanism by which VZV reactivation could lead to or accelerate the development of dementia is not known, there is evidence supporting the association between HZ and dementia[4,12–16]. Here, we observed that the incidence of dementia was higher in those diagnosed with HZ, regardless of RZV vaccination status. It is worth noting that this post-hoc analysis was limited as any report of HZ diagnosis, including those occurring both prior to and following the index date, were combined, and the analysis was not adjusted for differences in baseline cohort characteristics between the RZV-vaccinated and unvaccinated populations.

The results of this study are robust in a series of analyses addressing possible sources of bias. Much of the reduction in risk of dementia is unlikely to be explained by healthy vaccinee bias, as a 27% reduction in dementia risk was still observed among individuals who received two doses of RZV as compared to individuals who only received Tdap immunization during the accrual period. Further, evaluation of a composite NCO that should share similar potential sources of bias as the primary analysis had an aHR close to 1.00, indicating that residual confounding from unmeasured confounding such as lifestyle and healthcare seeking behavior is likely to be minimal. Additionally, the reduction in incidence of dementia was consistent across sensitivity analyses, including an analysis using an alternative dementia definition with stricter requirements, an analysis in which individuals were not censored if they received a dose of HZ vaccine, and an analysis excluding patients with index dates during the peak of the COVID-19 pandemic to address potential changes in healthcare seeking behaviors. The reduction in risk compared to unvaccinated individuals

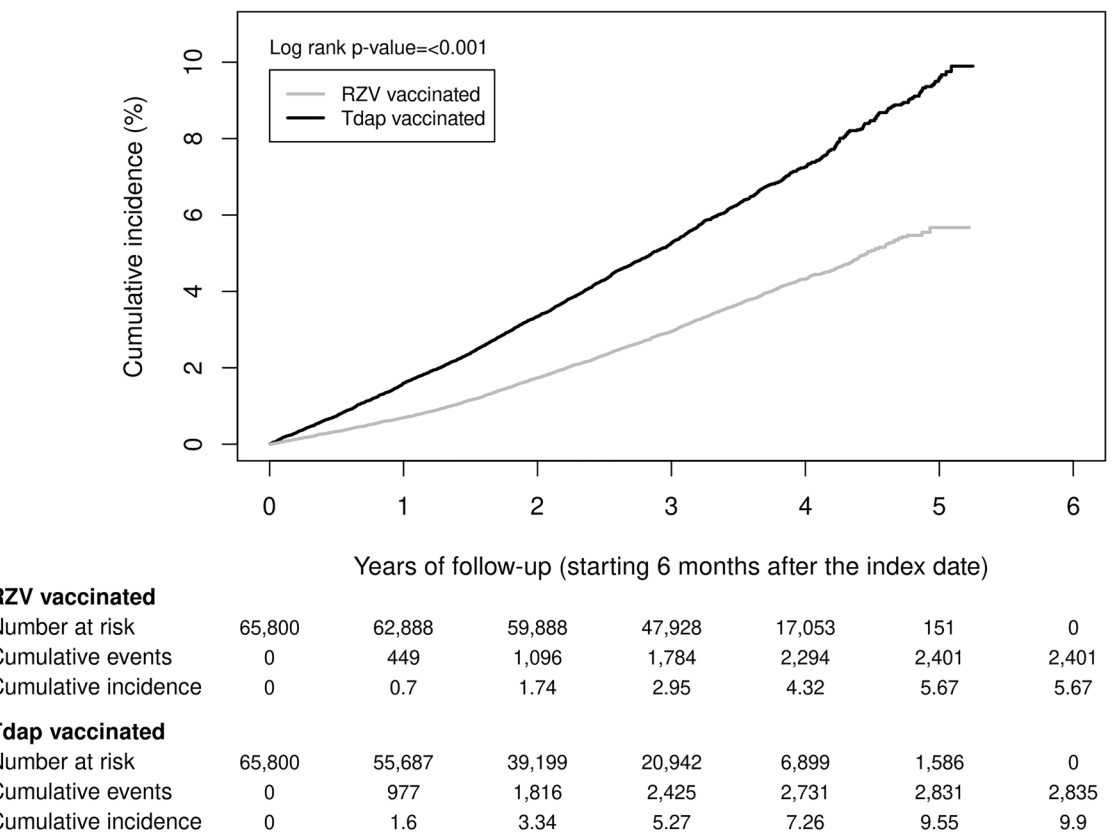

**RZV vaccinated**

| | | | | | | |
|---|---|---|---|---|---|---|
| Number at risk | 65,800 | 62,888 | 59,888 | 47,928 | 17,053 | 151 | 0 |
| Cumulative events | 0 | 449 | 1,096 | 1,784 | 2,294 | 2,401 | 2,401 |
| Cumulative incidence | 0 | 0.7 | 1.74 | 2.95 | 4.32 | 5.67 | 5.67 |

**Tdap vaccinated**

| | | | | | | |
|---|---|---|---|---|---|---|
| Number at risk | 65,800 | 55,687 | 39,199 | 20,942 | 6,899 | 1,586 | 0 |
| Cumulative events | 0 | 977 | 1,816 | 2,425 | 2,731 | 2,831 | 2,835 |
| Cumulative incidence | 0 | 1.6 | 3.34 | 5.27 | 7.26 | 9.55 | 9.9 |

**Fig. 5 | Cumulative incidence of dementia among two-dose RZV-vaccinated versus Tdap-vaccinated individuals.** RZV recombinant zoster vaccine, Tdap tetanus, diphtheria, and acellular pertussis vaccine.

| | Incidence per 1000 person-years (95% CI) | | aHR (95% CI)[a] | |
|---|---|---|---|---|
| Outcome | Vaccinated | Unvaccinated | | |
| Dementia | 10.74 (10.32–11.18) | 23.04 (22.61–23.48) | | 0.49 (0.46–0.51) |
| Any negative control outcome listed below: | 9.75 (9.34–10.16) | 10.06 (9.78–10.35) | | 0.94 (0.89–1.00)[d] |
| Wrist fracture[b] | 2.73 (2.52–2.96) | 2.26 (2.13–2.40) | | 1.15 (1.03–1.29) |
| Acute pancreatitis[c] | 1.69 (1.53–1.87) | 2.34 (2.21–2.48) | | 0.75 (0.66–0.85) |
| Appendicitis[c] | 0.88 (0.77–1.01) | 0.90 (0.82–0.99) | | 0.98 (0.81–1.18) |
| Acute cholecystitis[c] | 0.75 (0.65–0.87) | 1.11 (1.02–1.21) | | 0.70 (0.58–0.86) |
| Adhesive capsulitis of the shoulder[c] | 3.00 (2.78–3.23) | 2.69 (2.54–2.84) | | 1.05 (0.94–1.17) |
| Trigeminal neuralgia[c] | 0.84 (0.72–0.96) | 0.91 (0.83–1.00) | | 0.84 (0.70–1.01) |

0.0 0.2 0.4 0.6 0.8 1.0 1.2

**Fig. 6 | Risk of dementia and negative control outcomes among two-dose RZV-vaccinated versus unvaccinated individuals.** Incidence of dementia and negative control outcomes with 95% confidence intervals and adjusted hazard ratios with 95% confidence intervals calculated using a Cox regression model with IPTW are presented for two-dose RZV-vaccinated ($n = 65,800$) compared to unvaccinated ($n = 263,200$) individuals. [a]Adjusted for time-varying vaccination status (influenza, COVID-19, and other vaccines) besides IPTW. [b]Wrist fracture included as a negative control outcome recommended in Salmon DA, et al. J Infect Dis. 2023;11:1224-1226. [c]Composite negative control outcome modeled from Taquet M et al. Nature Medicine. 2024;30:2777-2781. [d]Upper bound of the confidence interval is 0.998. aHR adjusted hazard ratio, CI confidence interval, IPTW inverse probability of treatment weighting, n number, RZV recombinant zoster vaccine.

was smaller, but still significant, when the start to follow-up time was delayed to 1 year instead of 6 months after index date (aHR 0.60 vs 0.49). The difference based on start of follow-up may be at least partially attributable to healthy vaccinee bias as the reduction in risk compared to Tdap-vaccinated individuals after a 1-year start to follow-up was the same as that observed in the main Tdap-comparator analysis (aHR 0.73 vs 0.73).

Our study has several strengths, the first of which is size, diversity, and stability of our study cohort, which supports the generalizability of our findings. KPSC's high member retention (75% of members remain after >5 years) supports the longitudinal ascertainment of baseline characteristics and follow-up on outcomes[38]. Second, the comprehensive electronic healthcare record (EHR) system at KPSC enables accurate capture of RZV vaccination, dementia, and MCI diagnosis, along with other sociodemographic and clinical covariates[39]. Third, chart review of the dementia and MCI outcome definitions increases confidence in code-based identification of outcomes, as they were consistent with documented diagnoses in the EHR. Fourth, findings across demographic groups and subtypes of dementia were consistent, indicating these results are not limited to a specific subgroup. Fifth, the Tdap sensitivity analysis and composite NCO indicate that study results are robust.

This study also has several limitations. First, these results may not be generalizable to patients who receive care in different types of

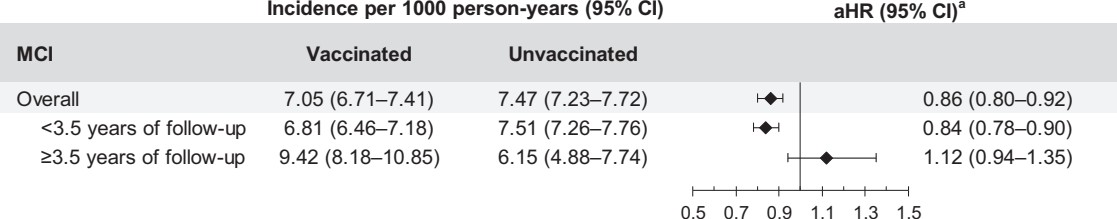

| MCI | Incidence per 1000 person-years (95% CI) | | aHR (95% CI)[a] |
| | Vaccinated | Unvaccinated | |
| --- | --- | --- | --- |
| Overall | 7.05 (6.71–7.41) | 7.47 (7.23–7.72) | 0.86 (0.80–0.92) |
| <3.5 years of follow-up | 6.81 (6.46–7.18) | 7.51 (7.26–7.76) | 0.84 (0.78–0.90) |
| ≥3.5 years of follow-up | 9.42 (8.18–10.85) | 6.15 (4.88–7.74) | 1.12 (0.94–1.35) |

**Fig. 7 | Risk of MCI among two-dose RZV-vaccinated versus unvaccinated individuals.** Incidence of MCI with 95% confidence intervals and adjusted hazard ratios with 95% confidence intervals calculated using a Cox regression model with IPTW are presented for individuals without MCI diagnosis prior to the start of follow-up in comparison of two-dose RZV-vaccinated (n = 64,744) to unvaccinated (n = 258,199) individuals. [a]Adjusted for time-varying vaccination status (influenza, COVID-19, and other vaccines) besides IPTW. aHR adjusted hazard ratio, CI confidence interval, IPTW inverse probability of treatment weighting, MCI mild cognitive impairment, n number, RZV recombinant zoster vaccine.

health systems in the US, including uninsured populations, or in other countries, or to those who received the second dose of RZV more than 6 months after the first. Second, it is possible that residual confounding may exist; however, the NCO results suggest that residual confounding is likely minimal. Additionally, an E-value of 3.50 indicates that a confounder not measured in this analysis would need to have a very strong relationship with both RZV vaccination and dementia (hazard ratio ≥3.5) to explain these findings as an artifact of confounding, which is unlikely given the wide variety of potential confounders adjusted for in this study. Third, dementia is a progressive disease, and the mean follow-up time after index date was 3.40 years in the RZV-vaccinated population, with a maximum follow-up time of 5.22 years. Ideally, this study would include a longer follow-up period as dementia develops over a long period of time; however, this study should still be representative of the data currently available on RZV, as it was only licensed in the US in 2017 and became available at KPSC in early 2018. Fourth, it is possible that there is differential misclassification of outcomes due to differences in healthcare seeking behaviors; however, healthcare utilization was included as a covariate in the analysis and reduction in dementia risk was still observed with RZV vaccination, even when compared to individuals who received Tdap vaccination during the accrual period. Fifth, the majority (63%) of unvaccinated individuals were censored due to receipt of RZV during the follow-up period. However, these individuals could subsequently contribute person-time to the RZV-vaccinated cohort if they otherwise met eligibility criteria during the accrual period, which should help to mitigate healthy vaccinee bias. Sixth, misclassification of dementia or MCI is possible due to the progressive nature of cognitive impairment, varied presentation in symptoms, and lack of consistent quantitative testing. Chart reviews demonstrated that the code-based identification of dementia and MCI had a high PPV; however, this study only identified cases that presented to healthcare providers and relied on the dates associated with coded dementia or MCI diagnosis, regardless of first presentation of symptoms. The underrepresentation of MCI cases in this study may be attributed to patients exhibiting milder symptoms, and who may not present for care for those symptoms. In conclusion, two doses of RZV were statistically significantly associated with a reduction in the risk of dementia in adults aged 65 years and older. After accounting for healthy vaccinee bias, RZV vaccination remained associated with a statistically significant lower risk of dementia. While additional research is needed to understand the potential underlying mechanisms, these results suggest that there may be additional benefits to RZV vaccination beyond prevention of HZ.

## Methods

### Study setting
KPSC is a large integrated health care system that includes 4.9 million members with diverse sociodemographic characteristics, reflecting the Southern California population[38,40]. Comprehensive patient health information, including demographics, diagnoses, vaccinations, and pharmacy records of KPSC members are stored in the EHR. Demographic information, including the patient's sex, is documented by healthcare providers during clinical encounters from available medical history and, if needed, patient self-report. KPSC is a prepaid healthcare system, and recommended vaccines, including RZV, are provided to members at no charge, incentivizing members to receive vaccines within the KPSC system. Additionally, vaccinations received outside of the health plan that are self-reported with appropriate documentation are also recorded in the EHR. The KPSC Institutional Review Board approved this study and waived the requirement for informed consent.

### Study population
Adults aged 65 years and older at the index date (defined under "Study design") with at least 1 year of continuous KPSC membership before the index date (allowing for a 31-day gap in membership) and until at least 6 months afterwards were eligible for inclusion. Individuals were excluded if they were diagnosed with dementia (by ICD-10 [International Classification of Diseases, Tenth Revision] code; Supplementary Table 17) or were prescribed any medication to treat dementia symptoms prior to or on the index date or within 6 months following the index date (Supplementary Table 18). Additionally, individuals were excluded if they received RZV or ZVL within 6 months following the index date, received the second dose of RZV less than 4 weeks after the first dose, or died within 6 months following the index date.

### Objectives
The primary objective of the study was to evaluate the association between two doses of RZV and reduction in risk of dementia in US adults aged 65 years and older (Fig. 1a). Secondary and exploratory objectives were to evaluate the association between two doses of RZV and reductions in (1) risk of dementia, by age, sex, race and ethnicity, ZVL status, and MCI status, (2) risk of dementia subtype (Alzheimer's disease, vascular dementia), (3) risk of dementia compared to individuals who received Tdap vaccine, and (4) risk of MCI, along with time to dementia diagnosis following incident MCI diagnosis during the follow-up period.

### Study design
We conducted a retrospective cohort study among KPSC members aged 65 years and older. The RZV-vaccinated group included adults who received two doses of RZV 4 weeks to 6 months apart between 01 April 2018 and 31 December 2020. Vaccinated individuals were 1:4 matched by age, sex, race, and ethnicity, and history of ZVL vaccination to unvaccinated individuals who had not received RZV as of the index date. The index date was defined as the date of the receipt of the second dose of RZV for vaccinated individuals and unvaccinated individuals were assigned the same index date as their vaccinated

matches. Follow-up time began 6 months after the index date to allow time for diagnosis of dementia cases that were present prior to vaccination but had not yet been diagnosed in the medical record. Follow-up lasted until receipt of an additional dose of HZ vaccine, termination of KPSC membership (allowing for a 31-day gap in membership), occurrence of an event of interest, death, or the end of the follow-up period (31 December 2023), whichever occurred first (Fig. 1b). Unvaccinated individuals who were matched to RZV-vaccinated individuals could also contribute person-time to the RZV-vaccinated cohort if they received two doses of RZV and otherwise met eligibility criteria; their unvaccinated person-time was censored upon receipt of the first dose of RZV.

A period of 1 year before the index date was used to define most baseline variables (e.g., comorbidities and healthcare utilization; Supplementary Methods). However, for some variables (e.g., dementia history, immunosuppressive conditions identified using diagnosis codes at KPSC), all available data from October 2015 onward were used, while for other variables (i.e., MCI history, ZVL history, HZ history, length of membership, immunosuppressive conditions identified using registry data), all available data were used. The study was conducted in accordance with the International Society for Pharmacoepidemiology Guidelines for Good Pharmacoepidemiology Practices[41].

## Study exposure and outcomes

The exposure of interest was the receipt of two doses of RZV 4 weeks to 6 months apart (Fig. 1c). RZV doses were identified using CVX (vaccine administered) code 187: zoster vaccine recombinant. The primary outcome was all-cause dementia, defined by ≥1 ICD-10 diagnosis code for dementia (Supplementary Table 17) from hospital, outpatient (including virtual), and ED settings during the study follow-up period (Fig. 1c). Secondary outcomes included Alzheimer's disease, vascular dementia, and MCI, defined by ≥1 ICD-10 diagnosis code (Supplementary Table 17) from hospital, outpatient (including virtual), and ED settings during the study follow-up period. Chart review was performed on a random sample of 100 records with coded dementia diagnoses and 100 records with coded MCI diagnoses, to assess whether the diagnosis codes aligned with the clinicians' documentation in the EHR. The PPV was calculated by dividing the number of confirmed cases by the number of reviewed cases.

## Statistical methods

**Primary analysis.** Baseline characteristics of the vaccinated and unvaccinated groups in the matched cohort were described. ASD was calculated to assess the balance of covariates, with an ASD < 0.1 considered a negligible difference[42].

Overall incidence rates of dementia were calculated by dividing the number of incident dementia cases by the total number of person-years; the 95% CI was calculated using SAS PROC GENMOD with a log link function assuming the incident dementia cases follow a Poisson distribution. The cumulative incidence of dementia was estimated by the Kaplan–Meier method. Vaccinated and unvaccinated cumulative incidence estimates were plotted, and the difference between two curves was tested by the log-rank test. The proportional hazards assumption was evaluated visually (Kaplan–Meier method) or by using an appropriate test (e.g., testing interactions of the exposure and survival time) if cumulative incidence curves crossed.

The incidence of dementia was compared between the vaccinated and unvaccinated groups, adjusting for potential confounders using an inverse probability of treatment weighting approach (IPTW)[43]. The probability of receiving two doses of RZV (i.e., propensity scores) as predicted by baseline covariates (Supplementary Table 19) was estimated using a logistic regression model (Supplementary Methods). Unadjusted and adjusted HRs and 95% CIs of dementia in the vaccinated and unvaccinated groups were estimated by two Cox proportional hazards regression models, one without IPTW and one with

IPTW (SAS PROC PHREG with a WEIGHT statement in which each observation in the input data set is weighted by the value of the weight variable). Additionally, weighted Cox regression models were adjusted for vaccinations (influenza vaccine, COVID-19 vaccine, or other vaccines) received during follow-up as time-varying covariates. Study conclusions were based on the adjusted (weighted) models (Supplementary Methods). Death was considered as a competing risk using cause-specific hazards models in which the competing events were treated as censored observations.

**Secondary and exploratory analyses.** Dementia incidence and aHRs in the RZV-vaccinated group and matched unvaccinated group stratified by age (65–69 years, 70–79 years, and ≥80 years) at index date, sex, race and ethnicity, ZVL status, and baseline MCI status were calculated with 95% CIs, using a similar method as for the primary analysis. Covariate balance within subgroups was assessed using the weights from the primary analysis; imbalanced variables (ASD > 0.1) were added to the adjusted models. If >20% of the variables were imbalanced, propensity scores were recalculated for the subgroup. Incidence and aHR of Alzheimer's disease and vascular dementia were calculated similarly to the primary analysis.

An analysis similar to the primary objective, using Tdap-vaccinated instead of unvaccinated individuals as a comparator, was conducted to mitigate bias due to "healthy vaccinee" effects. The Tdap vaccine was selected based on comparability to RZV in terms of indication for a similar patient population, as well as pattern of administration. The comparator group included individuals who received a dose of Tdap between 01 April 2018 and 31 December 2020, met the eligibility criteria, and did not receive a dose of RZV prior to the index date. The exposed RZV-vaccinated group remained the same as the primary analysis. Tdap recipients were matched 1:1 to RZV recipients on age and index date (date of second RZV dose and Tdap vaccination date) in the same year, or ±1 year if no matches were available in the same year. In a similar approach to the main analysis, Tdap-vaccinated individuals who were matched to RZV-vaccinated individuals could also contribute person-time to the RZV-vaccinated cohort if they received two doses of RZV and otherwise met eligibility criteria; their Tdap-vaccinated person-time was censored upon receipt of the first dose of RZV. The IPTW approach was used to account for covariates, including sex, race, and ethnicity, history of ZVL, and other potential confounders.

Among individuals without MCI diagnosis prior to the start of follow-up, incidence and aHR of MCI were calculated comparing the RZV-vaccinated group with the matched unvaccinated group. Due to violation of the proportional hazards assumption (Supplementary Fig. 3), the analysis was stratified into <3.5 years of follow-up and ≥3.5 years of follow-up. In addition, among the subgroup diagnosed with MCI and then dementia during follow-up, the time from MCI diagnosis to dementia diagnosis was described, stratified by RZV vaccination status.

In an exploratory unadjusted post-hoc analysis, incidence rates of dementia by HZ status (never diagnosed with HZ vs any HZ diagnosis prior to or during follow-up) were calculated for the RZV-vaccinated group and the matched unvaccinated group.

**Sensitivity analyses.** Individuals who receive RZV may differ from those who do not receive the vaccine in their risk of developing dementia and/or care-seeking behaviors for dementia. To assess whether the primary analysis of dementia may have been distorted by such confounders, a similar post-hoc analysis was conducted with a composite NCO comprised of acutely painful conditions not associated with dementia but likely to require medical attention[34,44]. The composite NCO consisted of six conditions: acute pancreatitis, appendicitis, acute cholecystitis, adhesive capsulitis of the shoulder, trigeminal neuralgia, and wrist fractures.

To further evaluate the potential for residual confounding, the E-value was calculated post-hoc for the primary analysis[45]. The E-value represents the minimum strength of association an unmeasured confounder would need to have to explain away the observed exposure-outcome association. A large E-value suggests that substantial unmeasured confounding would be needed to explain away the effect estimate. PPV was evaluated, and sensitivity analysis was conducted using an alternative outcome definition for dementia, requiring ≥2 dementia diagnoses or ≥1 dementia diagnosis with prescription medication for management of dementia symptoms (Supplementary Tables 17 and 18).

Due to the effect of the COVID-19 pandemic on healthcare-seeking behaviors, a sensitivity analysis for the primary objective was also conducted, excluding vaccinated individuals and their unvaccinated matches with an index date between 01 March 2020 and 31 December 2020.

An additional post-hoc sensitivity analysis was conducted in which receipt of HZ vaccine during follow-up was removed as a censoring criterion to account for possible behavioral changes in the unvaccinated population during the follow-up period. Individuals who were unvaccinated at the start of the study continued contributing person-time to the unvaccinated cohort (disregarding subsequent receipt of HZ vaccine during follow-up) until termination of KPSC membership, dementia diagnosis, death, or the end of follow-up (31 December 2023), whichever occurred first. Thus, individuals who originally contributed person-time to both the unvaccinated and RZV-vaccinated arms ($n = 25{,}987$) were not included in the RZV-vaccinated arm for this sensitivity analysis; their unvaccinated matches ($n = 103{,}948$) were also excluded.

An additional post-hoc sensitivity analysis was conducted in which the follow-up time began 1 year after the index date, instead of 6 months, to potentially (1) reduce healthy vaccinee bias as it is expected that this effect would become less pronounced further from vaccination, (2) reduce misclassification of cases with onset of dementia preceding the index date, and (3) allow a longer period for RZV vaccination to take effect on dementia outcomes.

**Reporting summary**

Further information on research design is available in the Nature Portfolio Reporting Summary linked to this article.

## Data availability

Individual-level data are not publicly available due to privacy concerns and protection of patient identities. Requests for aggregate-level data may be submitted to KPSC and are subject to review. De-identified aggregate-level data that support the findings of this study may be shared upon approval of a proposal and a signed data access agreement.

## Code availability

Epidemiological analyses were conducted using standard commands in SAS 9.4 (SAS Institute; Cary, North Carolina, USA).

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

## Acknowledgements

This work was supported and funded by GSK. Medical writing assistance for the manuscript and coordination were provided by Akkodis Belgium c/o GSK. The authors would like to thank the patients of Kaiser Permanente for their partnership. Their information, collected through our electronic health record system, leads to findings that help us improve care for our patients and can be shared with the larger community.

## Author contributions

E.R. is the guarantor of the work and accepts full responsibility for the integrity of the study and the decision to publish. E.R., L.S.S., L.Q., B.K.A., J.T., Y.L., E.C-Y., D.O., H.Y., C.K., and H.F.T. designed and conducted this study, including interpretation of data. L.Q., J.T., and Y.L. performed statistical analysis. P.P.M. and R.O.C. provided administrative, technical, and material support. E.R. drafted the manuscript. All authors critically revised the manuscript and contributed to the final drafting. The corresponding author attests that all listed authors meet authorship criteria and that no others meeting the criteria have been omitted.

## Competing interests

The authors declare the following competing interests: E.R., L.S.S., L.Q., B.K.A., Y.L., P.P.M., and H.F.T. received research funding from AstraZeneca. E.R., L.S.S., L.Q., B.K.A., J.T., Y.L., P.P.M., and H.F.T. received research funding from Moderna. ER and BKA received research funding from F2G, Inc. L.S.S., L.Q., and B.K.A. received research funding from Dynavax. B.K.A., J.T., and R.O.C. received research funding from Pfizer. B.K.A. received research funding from Genentech. E.R., L.S.S., L.Q., B.K.A., Y.L., P.P.M., and H.F.T. received research funding from GSK (unrelated to this study). E.C-Y., D.O., H.Y., and C.K. are employed by GSK and hold financial equities in GSK. The authors declare no other financial or non-financial interests.

## Additional information

Meeting(s) where the information has previously been presentedFindings from this study were presented at the Alzheimer's Association International Conference 2025.

