## [Peer Review file · Nature Communications]

Recombinant zoster vaccine is associated with a reduced risk of dementia

Corresponding Author: Dr Emily Rayens

Version 0:

Reviewer comments:

Reviewer #1

(Remarks to the Author)

This manuscript reports the results of a longitudinal study on recombinant zoster vaccine (RZV) and the risk of dementia among participants in the Kaiser Permanente Southern California database. Strengths of the investigation include the large sample size, the validation of outcome, the use of a vaccinated control group, and the extensive sensitivity analyses. Although the apparent protective effect may appear robust, there is a major source of potential bias that apparently has not been considered:

A very large proportion of unvaccinated individuals are censored because they received RZV during the period of follow-up. Although this proportion is not directly reported, it can be inferred from the data provided in Extended figure 1 -- for example, between years 2 and 3 after the index date, the number at risk declines by 20% in the vaccinated and by 57% in the unvaccinated, a difference almost certainly entirely due to incident RZV vaccination. This is problematic, because it is plausible that individuals remaining at risk are more likely to have cognitive decline and be at higher risk of dementia.

I cannot think of an easy fix of the potential bias above, but at the very least it should be acknowledged and the Authors should report the results of sensitivity analyses in which unvaccinated individuals continue to contribute to follow-up to the unvaccinated group until 31 December 2023, or until they are censored for other reasons. This would be a very conservative analyses, i.e. biased toward the null, because RZV vaccination could reduce risk of dementia in the unvaccinated group, but any remaining inverse association between RZV and dementia would be far more convincing.

Reviewer #2

(Remarks to the Author)

Thank you for the opportunity to review this study investigating the relationship between two doses of the recombinant Zoster Vaccine (RZV) and subsequent dementia incidence in a large sample of adults in Southern California. The study is well designed and methodologically rigorous. It will make an important contribution to the evidence base for assessing the potentially protective effects of Herpes Zoster (Shingles) vaccines against dementia - an area of substantial public health interest.

The study uses a retrospective matched cohort study design with inverse probability of treatment weighting to improve balance between groups. Of particular note is the extra effort to compare the RZV vaccinated group with a non-shingles vaccinated group (Tdap) and a negative control. These extra analyses help provide reassurance that the findings are unlikely to be explained purely by health vaccinee effects which have been limitations of prior studies.

I have only minor comments for the authors to consider:

1. It seems that one key difference between the RZV group and their matched controls was that the RZV groups must have interacted with a health service provider in the period 1 April 2018 and 31 Dec 2020, whereas the unvaccinated controls may not have had any interaction? Could this difference indicate more proactive health behavior among the RZV group? I note that the Tdap comparison does account for this but perhaps this difference could be discussed.
2. It was not clear why there was a six-month delay included between index date and follow-up commencement – this could be explained more clearly.
3. Page 8 – lines 202-203 – There is inconsistency with the methods section here. The results report: “There were 2,118 Tdap-vaccinated individuals (3.2%) who contributed person-time to both cohorts as they were vaccinated with RZV during follow-up”. This sounds like a control could cross over to become a case, but the methods (p18 lines 415-417) state that

“Unvaccinated matches who were later vaccinated with RZV contributed person-time up until they received the first dose of RZV and were then censored.”

4. Figure 1 – a large proportion of people (almost 25%) were excluded at the first step in the flow diagram – were these people with >6 months between RZV doses? It would be good to explain who this group is a bit further and consider whether this step could bias results in any way.

5. Figure 2 – there are differences in the covariates included within sub-group models which seems to be based on whether the weights achieved appropriate balance within each sub-group. Wouldn't it be more robust to recalculate the IPTW weights within each sub-group and include a consistent set of covariates?

6. There was no detailed discussion of why the Tdap vaccine was chosen as a comparator rather than any other type of vaccine. One issue with Tdap is that because it includes the tetanus booster, it is possible that it may be prompted by a wound or injury and so may be more likely to be delivered in an ED setting than a vaccine like RZV. This means it could represent a slightly different group of people rather than just controlling for the healthy vaccinee effect.

Version 1:

Reviewer comments:

Reviewer #1

(Remarks to the Author)

The Authors in the original manuscript addressed confounding by using inverse probability of treatment weighting (IPTW) that effectively made the group of vaccinee and unvaccinated individuals comparable at baseline. However, this comparability does not apply to the pre- and post-vaccination person-time in those individuals who were unvaccinated at baseline. Conceivably, this group is heterogeneous with respect to several potential confounders, and those who decided to get the zoster vaccine are likely to have a lower dementia risk than those who did not.

I still believe that ignoring this source of potential bias is not acceptable.

Allowing non-vaccinated individuals to contribute person-time to the vaccination group would be equivalent to emulating a trial in which individuals randomized to placebo are analyzed as part of the group randomized to active treatment once they get the vaccine. The fact that this was "by design", as stated by the authors, does not make the individuals who got the vaccine during the follow-up comparable to those who did not. The analysis that I suggested is to keep unvaccinated individuals in the unvaccinated group until the end of the follow-up, whether or not they received the vaccine -- i.e. these individuals should not contribute person-time to the vaccinated group. This analysis would emulate an intention to treat analysis in the hypothetical trial. The large proportion of individuals who "broke protocol" by choosing vaccination will attenuate any effect of the vaccine, but there will be no bias due to lack of comparability between vaccinated and non-vaccinated individuals.

Reviewer #2

(Remarks to the Author)

Thank you for the opportunity to re-review this paper. The authors have addressed the previous comments comprehensively and clarified aspects of the design that were raised by both reviewers. I have the following remaining comments:

1. It is very clear now from the authors' responses that the control group could contribute person time to the vaccinated group if they were censored due to a treatment switch (ie they later were vaccinated). What is still not clear is how the IPTW weighting accounted for this treatment switch? It appears that the matching and weighting process occurred based on the original inclusion criteria, but if a large proportion of participants were able to swap groups, then do the two groups remain balanced over time?

2. If this issue has not currently been addressed, it would be possible to undertake an "intention-to-treat" sensitivity analysis that censors cases when they switch treatment and this might provide an important conservative bound for the findings. It would also be possible to incorporate censoring weights into the study design which may also address this problem.

Version 2:

Reviewer comments:

Reviewer #2

(Remarks to the Author)

No further comments. The additional sensitivity test appears to address the previous concerns regarding potential group imbalances and increases the robustness of the conclusions.

We thank the reviewers for their expertise and for taking the time to review our manuscript and to provide thoughtful comments. Please see our point-by-point responses below.

Reviewer #1 (Remarks to the Author):

This manuscript reports the results of a longitudinal study on recombinant zoster vaccine (RZV) and the risk of dementia among participants in the Kaiser Permanente Southern California database. Strengths of the investigation include the large sample size, the validation of outcome, the use of a vaccinated control group, and the extensive sensitivity analyses. Although the apparent protective effect may appear robust, there is a major source of potential bias that apparently has not been considered:

A very large proportion of unvaccinated individuals are censored because they received RZV during the period of follow-up. Although this proportion is not directly reported, it can be inferred from the data provided in Extended figure 1 -- for example, between years 2 and 3 after the index date, the number at risk declines by 20% in the vaccinated and by 57% in the unvaccinated, a difference almost certainly entirely due to incident RZV vaccination. This is problematic, because it is plausible that individuals remaining at risk are more likely to have cognitive decline and be at higher risk of dementia.

I cannot think of an easy fix of the potential bias above, but at the very least it should be acknowledged and the Authors should report the results of sensitivity analyses in which unvaccinated individuals continue to contribute to follow-up to the unvaccinated group until 31 December 2023, or until they are censored for other reasons. This would be a very conservative analyses, i.e. biased toward the null, because RZV vaccination could reduce risk of dementia in the unvaccinated group, but any remaining inverse association between RZV and dementia would be far more convincing.

Response: A higher censoring rate in the unvaccinated group due to subsequent RZV vaccination is expected by design; 63% of unvaccinated individuals in this study were censored due to RZV vaccination. Unvaccinated individuals who were matched to RZV-

vaccinated individuals could also contribute person-time to the RZV-vaccinated cohort if they received two doses of RZV during the accrual period (01 April 2018–31 December 2020) and otherwise met eligibility criteria; their unvaccinated person-time was censored upon receipt of the first dose of RZV. Because such individuals were allowed to contribute both unvaccinated and vaccinated person-time to analyses, the vaccinated and unvaccinated groups were more comparable, which helped to mitigate healthy vaccinee bias. In addition, a 27% reduction in risk of dementia was still observed in the Tdap comparison analysis, which was included to specifically address potential healthy vaccinee bias.

We have concerns about the suggested sensitivity analysis, as it seems that it would introduce misclassification bias if vaccinated person-time is counted towards the unvaccinated population. Additionally, it seems that person-time (and any dementia cases occurring during it) could then be double-counted in both the vaccinated and unvaccinated groups.

In response to your feedback, we've added the following language to the limitations section in the discussion (line 360): "Fifth, the majority (63%) of unvaccinated individuals were censored due to receipt of RZV during the follow-up period. However, these individuals could subsequently contribute person-time to the RZV-vaccinated cohort if they otherwise met eligibility criteria during the accrual period which should help to mitigate healthy vaccinee bias."

Reviewer #2 (Remarks to the Author):

Thank you for the opportunity to review this study investigating the relationship between two doses of the recombinant Zoster Vaccine (RZV) and subsequent dementia incidence in a large sample of adults in Southern California. The study is well designed and methodologically rigorous. It will make an important contribution to the evidence base for assessing the potentially protective effects of Herpes Zoster (Shingles) vaccines against dementia - an area of substantial public health interest.

The study uses a retrospective matched cohort study design with inverse probability of treatment weighting to improve balance between groups. Of particular note is the extra effort to compare the RZV vaccinated group with a non-shingles vaccinated group (Tdap) and a negative control. These extra analyses help provide reassurance that the findings are unlikely to be explained purely by health vaccinee effects which have been limitations of prior studies.

I have only minor comments for the authors to consider:

1. It seems that one key difference between the RZV group and their matched controls

was that the RZV groups must have interacted with a health service provider in the period 1 April 2018 and 31 Dec 2020, whereas the unvaccinated controls may not have had any interaction? Could this difference indicate more proactive health behavior among the RZV group? I note that the Tdap comparison does account for this but perhaps this difference could be discussed.

Response: We considered healthcare utilization in the year prior to index date in our analysis to address differences in healthcare seeking behaviors in the vaccinated and unvaccinated cohorts. This included the number of outpatient [including virtual], Emergency Department, and inpatient encounters, as well as preventive care (line 427; Supplementary Methods). After weighting, healthcare utilization was similar between cohorts (ASD <0.1; Table 1). Additionally, as mentioned above, the Tdap comparator cohort would have needed to interact with a health service provider in the same period as the RZV-vaccinated group.

The following language is included in the limitations section in the discussion (lines 356-360): “Fourth, it is possible that there is differential misclassification of outcomes due to differences in healthcare seeking behaviors; however, healthcare utilization was included as a covariate in the analysis and reduction in dementia risk was still observed with RZV vaccination, even when compared to individuals who received Tdap vaccination during the accrual period.”

2. It was not clear why there was a six-month delay included between index date and follow-up commencement – this could be explained more clearly.

Response: A 6-month period after the index date was used because dementia can develop over a long period, and lag time can be used to help exclude dementia cases that were present prior to vaccination but had not yet been diagnosed in the medical record. The 6-month period is based on other studies evaluating the association between vaccines and dementia outcomes, most of which implement some lag time between index date and dementia outcome, from 90 days to 2 years.^{1,2}

The following language has been added to the methods (line 418): “Follow-up time began 6 months after the index date to allow time for diagnosis of dementia cases that were present prior to vaccination but had not yet been diagnosed in the medical record.”

3. Page 8 – lines 202-203 – There is inconsistency with the methods section here. The results report: “There were 2,118 Tdap-vaccinated individuals (3.2%) who contributed person-time to both cohorts as they were vaccinated with RZV during follow-up”. This sounds like a control could cross over to become a case, but the methods (p18 lines 415-417) state that “Unvaccinated matches who were later vaccinated with RZV

contributed person-time up until they received the first dose of RZV and were then censored.”

Response: As noted by the reviewer, unvaccinated individuals contributed person-time until the end of the study period or until they met censoring criteria, including RZV vaccination. However, if the previously unvaccinated individual was vaccinated with 2 RZV doses 4 weeks-6 months apart during the accrual period and met the eligibility criteria, then they also contributed person-time as an RZV-vaccinated individual.

The following language has been added in the methods (lines 423-426): “Unvaccinated individuals who were matched to RZV-vaccinated individuals could also contribute person-time to the RZV-vaccinated cohort if they received two doses of RZV and otherwise met eligibility criteria; their unvaccinated person-time was censored upon receipt of the first dose of RZV.”

In the Tdap comparison, the same RZV-vaccinated cohort was used as in the primary objective. The Tdap comparator group consisted of individuals who received Tdap during the accrual period and had no history of RZV prior to the index date. As reasoned with the unvaccinated individuals in the primary objective, if a Tdap-vaccinated comparator later received 2 doses of RZV during the accrual period and otherwise met eligibility criteria, they would contribute person-time to the RZV-vaccinated cohort as well.

The following language has been added to the methods (line 497): “In a similar approach to the main analysis, Tdap-vaccinated individuals who were matched to RZV-vaccinated individuals could also contribute person-time to the RZV-vaccinated cohort if they received two doses of RZV and otherwise met eligibility criteria; their Tdap-vaccinated person-time was censored upon receipt of the first dose of RZV.”

4. Figure 1 – a large proportion of people (almost 25%) were excluded at the first step in the flow diagram – were these people with >6 months between RZV doses? It would be good to explain who this group is a bit further and consider whether this step could bias results in any way.

Response: Yes, this excluded any individuals who received the second dose of RZV either <4 weeks (n=523) or >6 months (n=41,599) after the first dose. The primary objective of this study was to specifically evaluate the risk of dementia in individuals who received their RZV series 4 weeks to 6 months apart, compared to unvaccinated individuals. Although RZV should be administered as 2 doses separated by 2–6 months, ACIP guidance specifies that a second dose administered <4 weeks after the first dose should be repeated, but a second dose administered ≥4 weeks after the first dose does

not need to be repeated.³ Furthermore, due to variation in real-world practice, the second dose may be given earlier than 2 months after the first dose.

We do not anticipate bias in limiting the population to those who received the second dose within 6 months after the first dose, given that we took steps to control for confounding and healthy vaccinee effects. However, it can affect generalizability in that findings apply to those who receive the second dose within 6 months of the first dose, and this population could possibly be different from those who receive the second dose later. If we were to include individuals who received the second dose >6 months after the first dose, determining an appropriate cutoff date would be challenging, and we would still miss some individuals. We chose to evaluate the ACIP-recommended dose interval instead.

We have added the following language in the limitations section (line 343): “First, these results may not be generalizable to patients who receive care in different types of health systems in the US, including uninsured populations, or in other countries, or to those who received the second dose of RZV more than 6 months after the first.”

5. Figure 2 – there are differences in the covariates included within sub-group models which seems to be based on whether the weights achieved appropriate balance within each sub-group. Wouldn't it be more robust to recalculate the IPTW weights within each sub-group and include a consistent set of covariates?

Response: For each subgroup analysis, we assessed the covariate balance using the weights from the primary analysis and added imbalanced variables in the outcome models for IPTW analysis. This is a common approach when the number of subgroup analyses is moderate to large. We also followed the pre-specified analysis plan that if more than 20% of the variables were unbalanced, we would recalculate the treatment weights.

6. There was no detailed discussion of why the Tdap vaccine was chosen as a comparator rather than any other type of vaccine. One issue with Tdap is that because it includes the tetanus booster, it is possible that it may be prompted by a wound or injury and so may be more likely to be delivered in an ED setting than a vaccine like RZV. This means it could represent a slightly different group of people rather than just controlling for the healthy vaccinee effect.

Response: Tdap was chosen based on similarity to RZV in terms of indication for a similar population and pattern of administration. In general, adults who have never received Tdap should receive a single dose of Tdap, and a Td/Tdap booster is recommended every 10 years, minimizing the likelihood of multiple Tdap administrations

in an individual during the study period. For the Tdap-vaccinated cohort, a small portion of the population (4.7%) received their index vaccine in the ED setting.

We considered the pneumococcal vaccine as a potential comparator; however, we ultimately decided against it due to its age- and risk-based recommendations, and individuals could have received multiple pneumococcal vaccine doses of various formulations during the study period. To the former point, pneumococcal vaccine is recommended for adults <65 years with certain underlying medical conditions or other risk factors and is recommended for all adults ≥65 years.⁴ In addition, recommendations for pneumococcal vaccination in adults changed during the study period.

The following language has been added to the methods (line 490): “The Tdap vaccine was selected based on comparability to RZV in terms of indication for a similar patient population, as well as pattern of administration.”

REFERENCES

1. Wiemken, T.L., Salas, J., Morley, J.E., Hoft, D.F., Jacobs, C. & Scherrer, J.F. Comparison of rates of dementia among older adult recipients of two, one, or no vaccinations. *J Am Geriatr Soc* **70**, 1157-1168 (2022).
2. Douros, A., Ante, Z., Suissa, S. & Brassard, P. Common Vaccines and the Risk of Incident Dementia: A Population-based Cohort Study. *J Infect Dis* **227**, 1227-1236 (2023).
3. Dooling, K.L., *et al.* Recommendations of the Advisory Committee on Immunization Practices for Use of Herpes Zoster Vaccines. *MMWR Morb Mortal Wkly Rep* **67**, 103-108 (2018).
4. Kobayashi, M., *et al.* Pneumococcal Vaccine for Adults Aged ≥19 Years: Recommendations of the Advisory Committee on Immunization Practices, United States, 2023. *MMWR Recomm Rep* **72**, 1-39 (2023).

We thank the reviewers for their expertise and for taking the time to review our manuscript and to provide thoughtful comments. Please see our point-by-point responses below.

REVIEWER COMMENTS

Reviewer #1 (Remarks to the Author):

The Authors in the original manuscript addressed confounding by using inverse probability of treatment weighting (IPTW) that effectively made the group of vaccinated and unvaccinated individuals comparable at baseline. However, this comparability does not apply to the pre- and post-vaccination person-time in those individuals who were unvaccinated at baseline. Conceivably, this group is heterogeneous with respect to several potential confounders, and those who decided to get the zoster vaccine are likely to have a lower dementia risk than those who did not.

I still believe that ignoring this source of potential bias is not acceptable.

Allowing non-vaccinated individuals to contribute person-time to the vaccination group would be equivalent to emulating a trial in which individuals randomized to placebo are analyzed as part of the group randomized to active treatment once they get the vaccine. The fact that this was "by design", as stated by the authors, does not make the individuals who got the vaccine during the follow-up comparable to those who did not. The analysis that I suggested is to keep unvaccinated individuals in the unvaccinated group until the end of the follow-up, whether or not they received the vaccine -- i.e. these individuals should not contribute person-time to the vaccinated group. This analysis would emulate an intention to treat analysis in the hypothetical trial. The large proportion of individuals who "broke protocol" by choosing vaccination will attenuate any effect of the vaccine, but there will be no bias due to lack of comparability between vaccinated and non-vaccinated individuals.

Response: We have added the suggested sensitivity analysis. We observed a 40% (95% CI: 37%-44%) reduction in risk of dementia with this approach, which included vaccinated person-time in the "unvaccinated" cohort. As expected, the hazard ratio

moved closer to the null, but vaccination with two doses of RZV was still associated with a significant reduction in risk of dementia diagnosis.

The following language has been added in the results (line 273): “In a post-hoc sensitivity analysis in which individuals were not censored upon receipt of HZ vaccine during follow-up, there were 39,813 RZV-vaccinated individuals and 159,252 unvaccinated individuals (Supplementary Table 15), with characteristics similar to the primary cohort. There were 1,540 dementia cases (IR: 10.95 [95% CI: 10.42–11.51] per 1,000 PY) in the RZV-vaccinated cohort and 9,834 dementia cases (IR: 20.02 [95% CI: 19.62–20.42] per 1,000 PY) in the unvaccinated cohort (Supplementary Table 14). The aHR of two doses of RZV and dementia was 0.60 (95% CI: 0.56–0.63).”

Additional information is presented in the methods (line 544) and discussion (line 331).

Reviewer #2 (Remarks to the Author):

Thank you for the opportunity to re-review this paper. The authors have addressed the previous comments comprehensively and clarified aspects of the design that were raised by both reviewers. I have the following remaining comments:

1. It is very clear now from the authors' responses that the control group could contribute person time to the vaccinated group if they were censored due to a treatment switch (ie they later were vaccinated). What is still not clear is how the IPTW weighting accounted for this treatment switch? It appears that the matching and weighting process occurred based on the original inclusion criteria, but if a large proportion of participants were able to swap groups, then do the two groups remain balanced over time?

Response: We thank the reviewer for highlighting this point as it is a valid concern in this study design. We assessed covariate balance during the follow-up period and found that most covariates remained balanced. A few showed slight imbalances, with ASDs between 0.1 and 0.2. However, the hazard ratio estimates remained similar after adjusting for these covariates (data not shown).

2. If this issue has not currently been addressed, it would be possible to undertake an “intention-to-treat” sensitivity analysis that censors cases when they switch treatment and this might provide an important conservative bound for the findings. It would also be possible to incorporate censoring weights into the study design which may also address this problem.

Response: We have included the suggested sensitivity analysis, with an observed 40% reduction in risk of dementia associated with 2 doses of RZV. Changes are detailed in the response to Reviewer 1 above.